# Multi-sensor information fusion localization of rare-earth suspended permanent magnet maglev trains based on adaptive Kalman algorithm

Yiwei Xu[1,2], Kuangang Fan[1,2,3]*, Qian Hu[1,2], Haoqi Guo[1,2]

**1** School of Electrical Engineering and Automation, Jiangxi University of Science and Technology, Ganzhou, China, **2** Key Laboratory of Magnetic Levitation Technology in Jiangxi Province, Ganzhou, China, **3** Ganjiang Innovation Academy, Chinese Academy of Sciences, Ganzhou, China

* kuangangfriend@163.com

**Data Availability Statement:** All relevant data are within the paper and its Supporting information files.

## Abstract

Since the positioning accuracy of sensors degrades due to noise and environmental interference when a single sensor is used to localize a suspended rare-earth permanent magnetically levitated train, a multi-sensor information fusion method using multiple sensors and self-correcting weighting is proposed for permanent magnetic levitated train localization. A decay memory factor is introduced to reduce the weight of the influence of historical measurement data on the fusion estimation, thus enhancing the robustness of the fusion algorithm. The Kalman filtering results suffer from inaccuracy when process noise is present in the system. In this paper, we use a covariance adaptive scheme that replaces the prediction step of the Kalman filter with covariance. It uses the covariance adaptive scheme to search the posterior sequence online and reconstruct the prior error covariance. Since the process noise covariance is not used in the new adaptive scheme, the negative impact of the mismatch noise statistics is greatly reduced. Simulation and experimental results show that the use of multi-sensor information fusion and covariance adaptive Kalman algorithm has significant advantages in terms of adaptability, accuracy and simplicity.

## Introduction

The problems of traffic congestion and environmental pollution caused by population concentration due to rapid economic development and urbanization have become imminent. As a typical green, safe and efficient mode of transportation, maglev rail transit has great potential for development [1]. Magnetic levitation, as an advanced technology for rail transportation, will promote further development and application in transportation [2, 3]. At present, China, the US, Germany and Japan are the four most developed countries in maglev technology, and are clearly leading in the international arena with abundant research results [4–6]. Meanwhile, countries such as Switzerland, Canada, and South Korea are actively researching new maglev rail technology [7]. The positioning system of permanent magnet magnetic levitation train plays a very important role in the safety of train dispatching and train movement control.

**Funding:** This work was supported in part by the Jiangxi Provincial Natural Science Foundation (20232ACB202001), in part by the Central Guided Local Science and Technology Funding Project of the Science and Technology Department of Jiangxi Province (Cross-regional Cooperation, 20221ZDH04052), in part by the 03 Special Project and 5G Program of the Science and Technology Department of Jiangxi Province (No. 20193ABC03A058), in part by the Program of Qingjiang Excellent Young Talents in Jiangxi University of Science and Technology (JXUSTQJBJ2019004), in part by the Key Research and Development Plan of Ganzhou (industrial field) ([2019]60), in part by a grant from the Research Projects of Ganjiang Innovation Academy, Chinese Academy of Sciences (No.E255J001), and in part by the cultivation project of the State Key Laboratory of Green Development and High-Value Utilization of Ionic Rare-Earth Resources in Jiangxi Province (20194AFD44003).

**Competing interests:** The authors have declared that no competing interests exist.

Accurately and without delay detecting the speed and position of the permanent magnet maglev train at a certain moment is the primary condition to ensure the safe operation of the whole system. Therefore, the research on the positioning of permanent magnet magnetic levitation trains is the basis for the future development of magnetic levitation trains, which is necessary and urgency.

When using single sensor for positioning of permanent magnet maglev trains, the positioning accuracy is degraded due to the presence of noise and occlusion in the environment, therefore a multi-sensor information fusion method is employed to solve this problem. Multi-sensor fusion technology applies data fusion [8, 9] to target tracking [10], vehicle localization [11] and other fields [12–14], which solves some problems of low accuracy in many cases and has broad application prospects and great scientific value [15]. In order to meet the requirements for positioning of permanent magnet magnetic levitation trains, multiple sensors are generally installed on the maglev trains for data acquisition. And then positioning is performed based on the acquired information. Therefore, the positioning accuracy of permanent magnet maglev trains depends on the accuracy of the acquired information. However, the noise, electromagnetic interference between devices, and environmental factors present in the practical application lead to random errors or mistakes in the sensors in a multi-sensor system [16]. This can cause bias or even distortion in the measurement results and ultimately lead to inaccurate positioning of the permanent magnet maglev train. Weighted fusion is a method for optimal data fusion by assigning a weighting factor to each sensor [17, 18], which produces optimal unbiased fusion results that minimize the mean square error fusion result without any a priori knowledge of the system and observation noise [19, 20]. Indeed, the performance of weighted fusion depends largely on the distribution of weights. If the weight distribution is not reasonable, it may not be able to significantly improve the accuracy and reliability of the system. Therefore, proper distribution of weights is an important factor in achieving high accuracy estimation in weighted fusion.

The Kalman filter has been widely used as an optimal state estimator in navigation, target tracking [21, 22]and control [23]. The optimality of the Kalman filter depends heavily on the a priori knowledge of the noise statistics [24]. The use of incorrect prior statistics may lead to significant estimation errors and even filter divergence [25]. Nevertheless, how to determine the process noise and the covariance of the measurement noise is a major hurdle in practice [26]. Therefore, it is rather meaningful to specially investigate the filtering problems with inaccurate or mismatched process noise covariance [27]. Adaptive techniques using covariance matching, correlation, maximum likelihood and Bayesian methods in combination with Kalman filtering are one of the common approaches to solve this problem [18]. The Sage-Husa adaptive Kalman filter is a covariance matching method that recursively estimates the noise statistics based on the maximum posterior criterion [28, 29]. The innovation-based adaptive Kalman filter (IAKF) is a maximum likelihood method that estimates the noise covariance matrix based on the fact that the innovation sequence of the Kalman filter is a white process [30, 31]. The multi-model adaptive Kalman filter is an approximation of the Bayesian approach that solves the problem of model uncertainty by combining Kalman filters of different models into a group [32, 33]. Kalman algorithm is more common in the previous studies of maglev train operation. In [34], the Kalman algorithm is used to filter the gap signal and obtain more ideal gap data, but the dynamic instability of this method is more obvious. In [35], Kalman technique is used to fuse multi-rate data with acceleration and displacement measurements having different sampling frequencies, and after numerical simulation and analysis, the effectiveness of the method is proved. In [36], an adaptive Kalman filtering algorithm based on the change of speed information is added into the operating speed sensor of low and medium-speed maglev trains to reduce the positioning error of maglev trains.

At present, China has researched different types of high-speed magnetic levitation technology [37, 38]. In 2014, Jiangxi University of Science and Technology proposed a new type of efficient and intelligent permanent magnetic levitation rail transportation system. In September 2019, a 60-meter-long permanent magnet magnetic levitation rail transit system technology verification line (Fig 1) was completed, marking the creation of a safe, convenient and efficient low to medium speed, low to medium capacity rail transit system. The current positioning method used in the technology verification line of the permanent magnet magnetic suspension rail system is a cross induction loop, which has the disadvantages of difficult installation, inconvenient maintenance and high cost. In response to these shortcomings, multiple sensors are used to locate the suspended rare earth permanent magnet maglev train, achieving multi-sensor information fusion positioning of the permanent magnet maglev train. This method has low positioning cost and wide application range. Accompanied by the rapid improvement of sensor technology, the use of multi-sensor fusion positioning can more effectively improve the effect of train positioning, not only to improve the stability and fault tolerance of the system, but also to ensure the accuracy of the position in the time and space range. At the same time, in order to make the further development of suspended rare-earth permanent magnet magnetically levitated trains, experts and scholars should study the positioning technology to provide the powerful theoretical support and detailed supporting materials for the subsequent commercial promotion and application of suspended rare-earth permanent magnetically levitated trains.

In this paper, a self-correcting weighted fusion estimation algorithm is first designed. A decay memory factor is introduced to reduce the weight of the influence of historical measurement data on the fusion estimation in order to fuse the information from multiple sensors of a permanent magnet maglev train. Second, a new covariance adaptive Kalman algorithm (CAKF) is used to adapt the error covariance online by prior values. Since CAKF calibrates the a priori error covariance directly through online feedback from random sequences, the

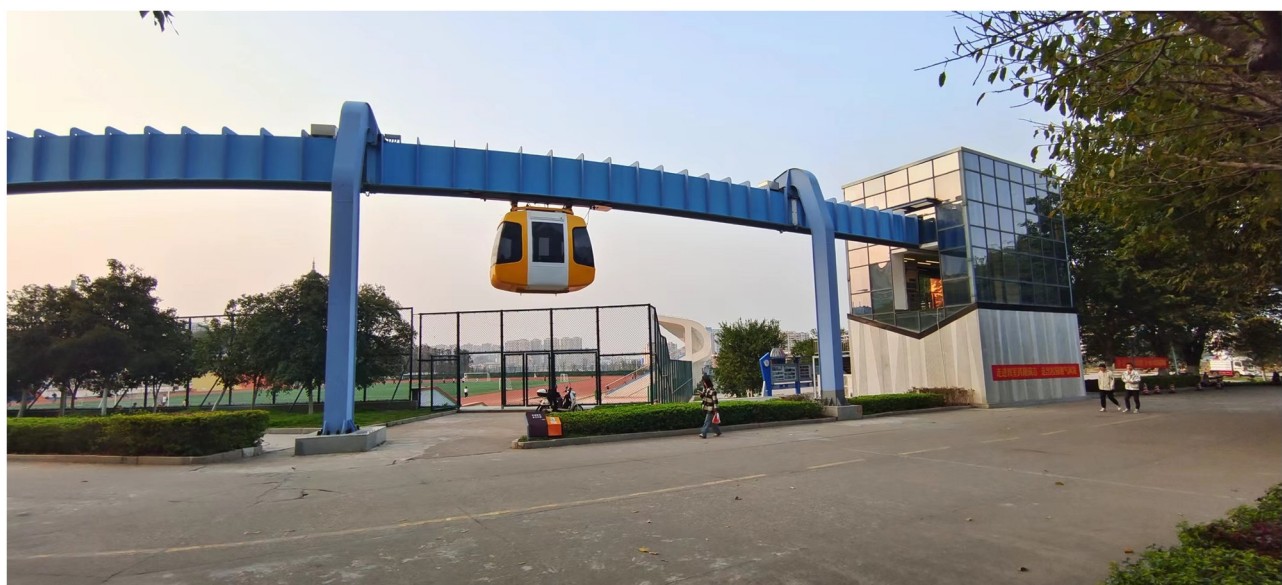

**Fig 1. Permanent magnet magnetic levitation rail transit system technology verification line.** Reprinted from [Jiangxi University of Science and Technology Permanent Magnet Magnetic Levitation Railway Transportation System Technology Validation Line] under a CC BY license, with permission from [Jiangxi University of Science and Technology], original copyright [2019].

filtering effect does not depend on the exact process noise covariance. Simulation and experimental results show that CAKF has a smaller root-mean-square error(RMSE).

## Model building

### Self-correcting weighted fusion multi-sensor model

The formula for the weighted fusion algorithm [39] is shown in Eq (1).

$$y(k) = \sum_{i=1}^{M} [\omega_i(k)z_i(k)] \tag{1}$$

Where $z_i(k)$ is the measurement value of the $i$th sensor at time $k$, $y(k)$ is the true state value of the observed object at time $k$, $\omega_i(k)$ is the weight of the $i$th sensor at time $k$, and $M$ is the number of sensors in the multi-sensor system.

The weight value of the distributed fusion algorithm determines the fusion accuracy. The conventional weight calculation method is based on the cumulative deviation of all historical measurement signals. It obtains the weight of each sensor dynamically based on the principle of least squares. This not only increases the storage volume of sensor measurement data, but also increases the influence weight of historical measurement signals. To address the above problems, an improved weight calculation method is designed by introducing a decay memory factor. The specific process is as follows:

- Step 1. Calculate the center point $\bar{z}(k)$ of the sensor's position at each moment.

$$\bar{z}(k) = \frac{1}{M} \sum_{i=1}^{M} z_i(k) \tag{2}$$

- Step 2. Calculate the deviation $\Delta z_i(k)$ between the measured value of each sensor and the center point.

$$\Delta z_i(k) = z_i(k) - \bar{z}(k) \tag{3}$$

- Step 3. Calculate the sum of the deviation and the squared deviation of the decay memory of each sensor.

$$S_{i1}(k) = \tau S_{i1}(k-1) + (1-\tau)\Delta z_i(k) \tag{4}$$

$$S_{i2}(k) = \tau S_{i2}(k-1) + (1-\tau)\Delta z_i^2(k) \tag{5}$$

Where, $S_{i1}(k)$ is the sum of the asymptotic memory deviations of the $i$th sensor at time $k$th moment, $S_{i2}(k)$ is the sum of the squared asymptotic memory deviations of the $i$th sensor at time $k$th moment, $\tau$ has a value range of [0, 1].

- Step 4. Calculate the average value $\Delta\bar{Z}_i(k)$ of the deviation for each sensor.

$$\Delta\bar{Z}_i(k) = S_{i1}(k)/M \tag{6}$$

- Step 5. Calculate the average value $\sigma_i(k)$ of the deviation for each sensor.

$$\sigma_i(k) = \left[ \frac{S_{i2}(k) - 2\Delta\bar{Z}_i(k)S_{i1}(k) + k\Delta\bar{Z}_i^2(k)}{k-1} \right]^{1/2} \tag{7}$$

- Step 6. Calculate the weight $\omega_i(k)$ of each sensor.

$$\omega_i(k) = \left[ \sigma_i^2(k)\sum_{i=1}^{M} \frac{1}{\sigma_i^2(k)} \right]^{-1} \tag{8}$$

The weights of each sensor are calculated according to the above method, and Eq (1) is used to calculate the final fusion estimation result at the $k$th moment.

## CAKF model

Consider the discrete time stochastic system as shown by the state-space model.

$$x_k = Ax_{k-1} + Bw_{k-1} \tag{9}$$

$$y_k = Cx_k + v_k \tag{10}$$

Where $k$ is the discrete time index, $x_k$ denotes the system state, $y_k$ is the measurement vector, $A$ is a known constant state transformation, $B$ is the input transformation, and $C$ is the measurement matrix. Both $w_k$ with covariance $Q$ and $v_k$ with covariance $R$ are mutually independent zero-mean Gaussian white noise.

Given the estimated value $\hat{x}_{k-1}^-$ and the measured value $y_k$, the Kalman filter outputs the optimal least squares estimate of the true state $x_k$ of instant $k$. The estimation of the Kalman is shown in Eqs (11)–(15)

$$\hat{x}_k^- = A\hat{x}_{k-1}^- \tag{11}$$

$$P_k^- = AP_{k-1}^-A^T + BQB^T \tag{12}$$

$$\hat{x}_k = \hat{x}_k^- + K_k(y_k - C\hat{x}_k^-) \tag{13}$$

$$K_k = P_k^-C^T(CP_k^-C^T + R)^{-1} \tag{14}$$

$$P_k = P_k^- - K_kCP_k^- \tag{15}$$

Where $\hat{x}_k^-$ and $\hat{x}_k$ denote the prior and posterior estimates of $x_k$ states, respectively, $P_k^-$ is the prior error covariance. $K_k$ is the Kalman gain matrix. For the optimal linear filter, the innovative sequence $e_k = y_k - C\hat{x}_k^-$ is a $\Theta_k = CP_k^-C^T$ Gaussian white noise with covariance, called the innovative of the optimal filter.

Since the focus of this paper is on the effect of the unknown process noise $Q$ on the results, the value of the measurement noise covariance $R$ is assumed to be completely known, which was shown to be feasible in the literature [40–42].

In Kalman filter, $Q$ is used in Eq (12). Therefore, in the new adaptive scheme, Eq (12) is replaced by an online search for the prior error covariance of the posterior sequence through

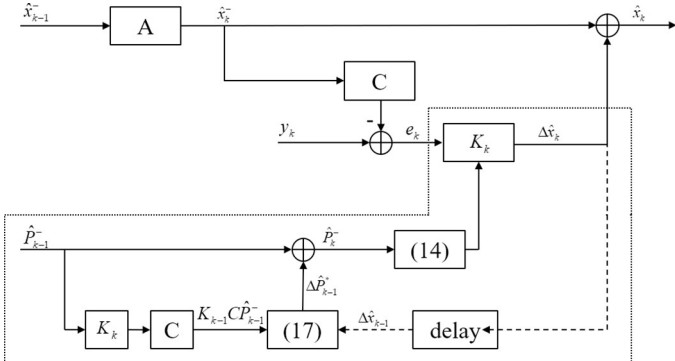

Covariance adaption scheme

**Fig 2. Schematic diagram of the CAKF.**

feedback.

$$\hat{P}_k^- = \hat{P}_{k-1}^- + \Delta\hat{P}_{k-1}^* \tag{16}$$

$$\Delta\hat{P}_{k-1}^* = (\Delta\hat{x}_{k-1}\Delta\hat{x}_{k-1}^T - K_{k-1}C\hat{P}_{k-1|k-2})/(k-1) \tag{17}$$

Where $\hat{P}_k^-$ is the estimate of the prior error covariance, $\Delta\hat{P}_{k-1}^*$ is the key feedback adaptation term, and $\Delta\hat{x}_{k-1}$ is the vector that is derived by subtracting the a priori estimate from the posteriori estimate.

$$\Delta\hat{x}_j = \hat{x}_j - \hat{x}_j^-, j = 1, 2, \ldots, k-1 \tag{18}$$

As shown in Fig 2, the $\Delta\hat{P}_{k-1}^*$ is first computed in Eq (17) using $\Delta\hat{x}_{k-1}$ and $K_{k-1}C\hat{P}_{k-1}^-$, and then the $\hat{P}_k^-$ is obtained by calibrating the final $\Delta\hat{P}_{k-1}^*$ with the feedback term $\hat{P}_{k-1}^-$. By comparing the mathematical expressions of Eqs (16) and (17) with (12) of the Kalman filter, the characteristics of CAKF scheme are as follows:

- The computational effort of the new scheme's method is approximately the same as that of Kalman's algorithm and does not require $Q$. This naturally relaxes the prior and accuracy constraints of Kalman theory on the covariance $Q$.

- The new scheme utilizes the posterior sequence and the feedback channel. Thus the posterior information in the sequence allows the timely adjustment of the prior at the next moment.

## Derivation of the covariance adaptive scheme

Assuming that $\varepsilon_k = \{e_1, e_2, \ldots e_{k-1}\}$ denotes the set of historical innovation sequences up to moment $k$. $\hat{P}_k^-$ is the unknown constant $P_k^-$ based on the estimation of $\varepsilon_k$, then

$$L(\hat{P}_k^-) = \ln p(\varepsilon_k|\hat{P}_k^-) = \ln\prod_{j=1}^{k-1} p(e_j|\hat{P}_k^-) = \sum_{j=1}^{k-1} \ln p(e_j|\hat{P}_k^-) \tag{19}$$

where $p(\varepsilon_k|\hat{P}_k^-)$ is the probability density of the set $\varepsilon_k$ and $p(e_j|\hat{P}_k^-)$ is the set density of the

Gaussian sequence $e_j$ conditioned to be $\hat{P}_k^-$.

$$p(e_j|\hat{P}_k^-) = p(e_j|\hat{P}_j^-) = \frac{1}{\sqrt{(2\pi)^m \det(\Theta_j)}} \exp\left(-\frac{1}{2}e_j^T \Theta_j^{-1} e_j\right) \tag{20}$$

Where $\det(\cdot)$ is the determinant operator, $j = 1, 2, \ldots, k-1$. $L(\hat{P}_k^-)$ is a scalar, while the covariance $\hat{P}_k^-$ is a symmetric matrix of $n \times n$, the derivative $M_k = \partial L(\hat{P}_k^-)/\partial \hat{P}_k^-$ is also a symmetric matrix of $n \times n$.

$$
\begin{aligned}
M_k^{s,t} &= -\frac{1}{2}tr\{\sum_{j=1}^{k-1}\left[(\Theta_j^{-1} - \Theta_j^{-1}e_je_j^T\Theta_j^{-1})\frac{\partial\Theta_j}{\partial(\hat{P}_k^-)^{s,t}}\right]\} \\
&= -\frac{1}{2}tr\{\sum_{j=1}^{k-1}\left[(\Theta_j^{-1} - \Theta_j^{-1}e_je_j^T\Theta_j^{-1})C\frac{\partial P_j^-}{\partial(\hat{P}_k^-)^{s,t}}C^T\right]\}
\end{aligned}
\tag{21}
$$

where $(\hat{P}_k^-)^{s,t}$ and $M_k^{s,t}$ are the $s$th row and the $t$th column of $\hat{P}_k^-$ and $M_k$ with $1 \le s \le n$, $1 \le t \le n$, respectively. Then, according to mathematical knowledge, the maximum value is obtained by setting $M_k^{s,t} = 0$.

$$tr\{\sum_{j=1}^{k-1}\left[C^T(\Theta_j^{-1} - \Theta_j^{-1}e_je_j^T\Theta_j^{-1})C\frac{\partial P_j^-}{\partial(\hat{P}_k^-)^{s,t}}\right]\} = 0 \tag{22}$$

The approximation of $P_k^-$ is constant and $\partial P_j^-/\partial(\hat{P}_k^-)^{s,t} = \partial \hat{P}_k^-/\partial(\hat{P}_k^-)^{s,t}$ is 0 except for the $s$th row and the element in column $t$, which is 1. Actually, $s$ and $t$ can be any value of $1, 2, \ldots, n$. (22) can be expressed into

$$\sum_{j=1}^{k-1}[C^T(\Theta_j^{-1} - \Theta_j^{-1}e_je_j^T\Theta_j^{-1})C] = 0_{n\times n} \tag{23}$$

Multiplying $P_k^- = P_j^-$ before and after the equation, we can obtain Eq (24).

$$\sum_{j=1}^{k-1}[P_j^-C^T(\Theta_j^{-1} - \Theta_j^{-1}e_je_j^T\Theta_j^{-1})CP_j^-] = 0_{n\times n} \tag{24}$$

Rewriting the above equation with Eq (14) and $\Theta_k = CP_k^-C^T + R$, Eq (25) was obtained.

$$\sum_{j=1}^{k-1}(K_jCP_j^- - K_je_je_j^TK_j^T) = 0_{n\times n} \tag{25}$$

Combining (13), (15), (17), $\Delta\hat{x}_j = K_je_j$ and $K_jCP_j^- = P_j^- - P_j$, we can obtain Eq (26).

$$\sum_{j=1}^{k-1}(P_j^- - P_j - \Delta\hat{x}_j\Delta\hat{x}_j^T) = 0_{n\times n} \tag{26}$$

And then

$$\sum_{j=1}^{k-1} P_j^- = \sum_{j=1}^{k-1} (P_j + \Delta \hat{x}_j \Delta \hat{x}_j^T) \tag{27}$$

Assuming that the approximation is $P_k^- = P_j^-, j < k$, the unknown $P_k^-$ at moment $k$ can be approximated by calculating $P_j^-$ averaged over all historical moments, so the estimated covariance $\hat{P}_k^-$ can be obtained by the following equation.

$$\hat{P}_k^- = \frac{1}{k-1} \sum_{j=1}^{k-1} P_j^- = \frac{1}{k-1} \sum_{j=1}^{k-1} (P_j + \Delta \hat{x}_j \Delta \hat{x}_j^T) \tag{28}$$

Similarly, for the previous moment of $k-1$,

$$\hat{P}_{k-1}^- = \frac{1}{k-2} \sum_{j=1}^{k-2} P_j^- = \frac{1}{k-2} \sum_{j=1}^{k-2} (P_j + \Delta \hat{x}_j \Delta \hat{x}_j^T) \tag{29}$$

By analogy, it can be written as Eq (30).

$$\hat{P}_k^- = \frac{k-2}{k-1} \hat{P}_{k-1}^- + \frac{1}{k-1} (P_{k-1} + \Delta \hat{x}_{k-1} \Delta \hat{x}_{k-1}^T) \tag{30}$$

In the new adaptive scheme, $P_{k-1}^-$ is denoted by $\hat{P}_{k-1}^-$ and combined with Eq (14), $P_{k-1}^- = \hat{P}_{k-1}^- - K_{k-1} C \hat{P}_{k-1}^-$ can be obtained, so the following equation can be obtained.

$$\hat{P}_k^- = \hat{P}_{k-1}^- + \frac{1}{k-1} (\Delta \hat{x}_{k-1} \Delta \hat{x}_{k-1}^T - K_{k-1} C \hat{P}_{k-1}^-) \tag{31}$$

## Simulation

In order to verify the fault tolerance performance and fusion accuracy of the self-correcting weighted fusion algorithm and the adaptive Kalman fusion algorithm, the designed algorithm is used for the fusion design of three sensors with random intermittent noise. In the simulation experiment of this paper, the designed permanent magnet maglev train point object does CA accelerated motion with acceleration of $1 m/s^2$ for 100 seconds, CV uniform motion for 100 seconds, and CA accelerated motion with acceleration of $-1 m/s^2$ for 100 seconds. The sampling time is $T = 1s$. The state vector is positioned as the position and speed of the permanent magnet maglev train, and the sensor collects the position of the point object according to Eq (10).

The target states are position and velocity, $X = [x, \dot{x}]^T$, and the CV model and CA model are Eqs (32) and (33), respectively.

$$X_k = \begin{bmatrix} 0 & 1 \\ 0 & 0 \end{bmatrix} X_{k-1} + \begin{bmatrix} T^2/2 \\ T \end{bmatrix} W_{k-1} \tag{32}$$

$$X_k = \begin{bmatrix} 1 & T & T^2/2 \\ 0 & 1 & T \\ 0 & 0 & 1 \end{bmatrix} X_{k-1} + \begin{bmatrix} T^2/2 \\ T \\ 1 \end{bmatrix} W_{k-1} \tag{33}$$

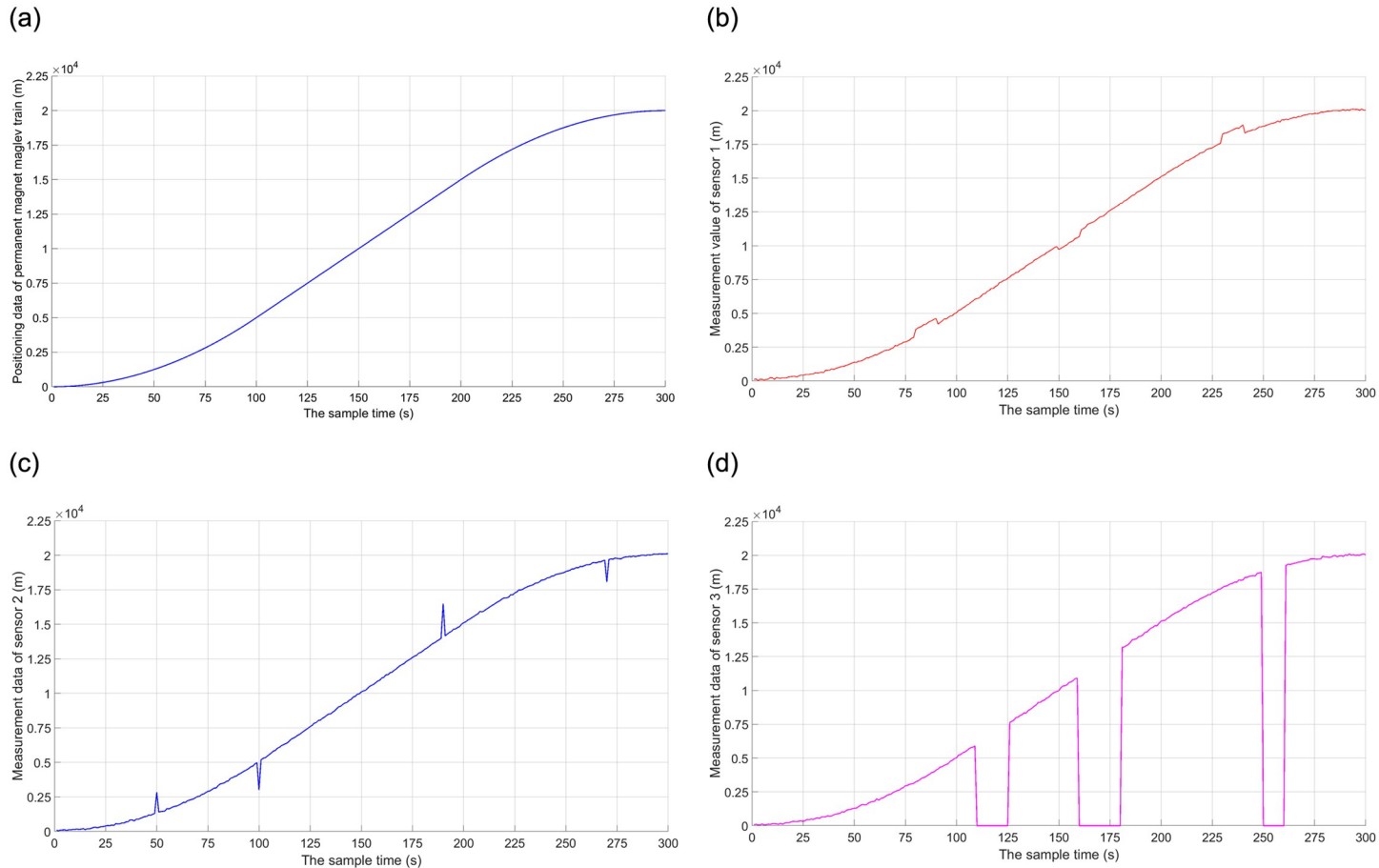

**Fig 3. Ideal state and measured values of three sensors.** (a) Ideal state; (b) Measured value of sensor 1; (c) Measured value of sensor 2; (d) Measured value of sensor 3.

The main situations where sensor data errors occur include sensor failure, sensor deviation, and significant sudden changes in sensor failure. In a sensor system, each sensor measures different results for information at the same location, and each sensor generates different types of data errors or errors, and at different times. In the simulation experiment, the measurement results after simulating the added noise and error for each sensor are shown in Fig 3.

As can be seen in Fig 3, the measurement signal of sensor 1 contains not only measurement noise but also a random drift signal with small amplitude. The measurement result of sensor 2 contains not only the measurement noise but also the pulse error signal. The signal has a large random amplitude, but a short duration. The measurement signal of sensor 3 contains measurement noise, and there also exist random signal masking and failure to receive the signal.

The three sensor signals are fused using the designed self-correcting weighted multi-sensor fusion algorithm for data fusion and compared with the results of the ideal state. As can be seen in Fig 4, the self-correcting weighted multi-sensor fusion algorithm is unaffected by erroneous data in the case of erroneous sensor measurements, distorted measurement drift and interrupted sensor signals. The fusion results are closer to the real positioning data. This indicates that the self-correcting weighted multi-sensor fusion algorithm is less affected by

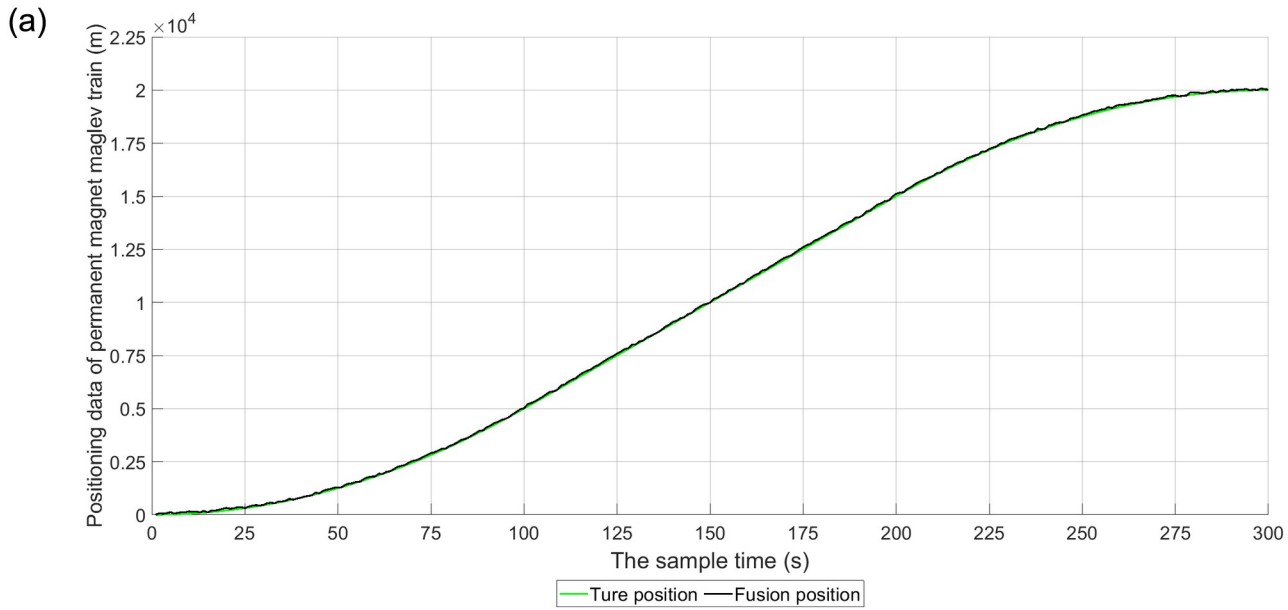

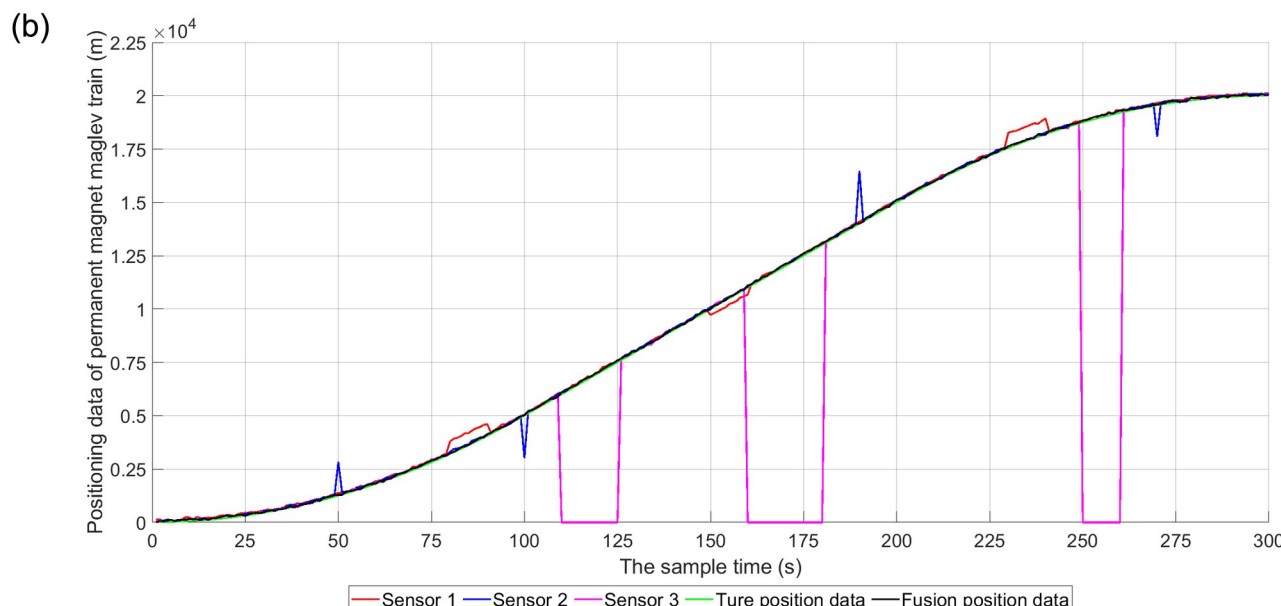

**Fig 4. Self-correcting weighted multi-sensor fusion results.** (a) Self-correcting weighted multi-sensor fusion localization results; (b) Comparison of multi-sensor positioning data.

faulty signals, has better fault tolerance performance, better robustness, and higher fusion accuracy.

In order to study the performance of Kalman, IAKF, and CAKF on different $Q$ values, simulation experiments with $Q$ values ranging from 0.1 to 10 were added. Assume that $R$ is known, so the effect of $R$ on the results is not considered anymore. To evaluate the accuracy,

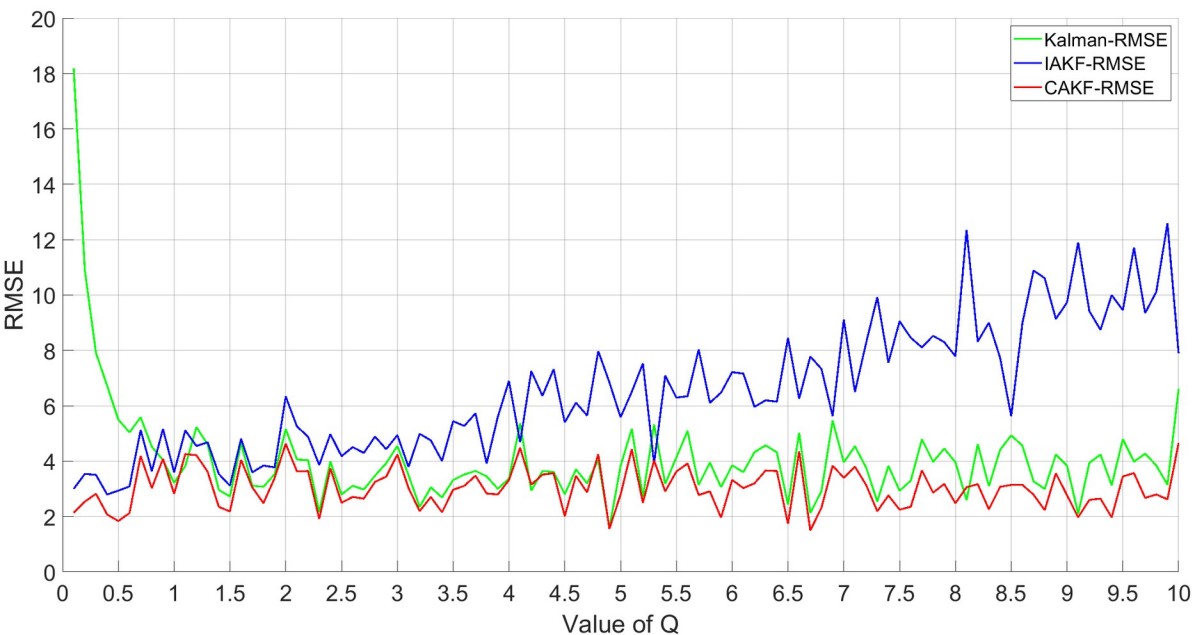

**Fig 5. RMSE results for different $Q$.**

the root mean square error is chosen as the performance metric, as shown in Eq (34).

$$RMSE = \sqrt{\frac{\sum_{k=1}^{N} (y_k - \hat{x}_k)^2}{N}} \qquad (34)$$

Based on the results in Fig 5, it is clear that: (1) Kalman algorithm shows great adaptability when the initial value of $Q$ is large, but when the initial value of $Q$ is small, the RMSE of Kalman algorithm is large and cannot meet the application requirements. (2) IAKF algorithm can obtain high fusion accuracy when the initial value of $Q$ is small. However, as the initial value of $Q$ increases, the RMSE of IAKF algorithm also increases, and the positioning accuracy decreases, which does not guarantee the adaptivity and cannot meet the needs of fusion. (3) The positioning accuracy of CAKF is significantly better than the Kalman algorithm when the initial value of $Q$ is small, and the value of RMSE is also smaller than that of the IAKF algorithm. At larger initial values of $Q$, the filtering effect is significantly better than the IAKF algorithm, and the error of CAKF is reduced compared with the Kalman algorithm. In other words, CAKF has better adaptability in positioning accuracy for different $Q$ initial values, and the new covariance adaptive scheme relaxes the constraints of Kalman theory on the previous exact $Q$.

To investigate the performance of the above method for different noise intensities, Gaussian noise with a mean of 0.1–3 and a variance of 1–30 was added to the measured signals fused by the self-correcting weighted multi-sensor fusion algorithm, and the simulation results are shown in Fig 6.

The simulation results show that: (1) for the traditional Kalman algorithm, both the IAKF algorithm and CAKF have better positioning effects than the Kalman algorithm. (2) The RMSE of CAKF is almost always smaller than the RMSE of the IAKF algorithm. This indicates that CAKF has better adaptability to noise of different intensities than other methods, and the new covariance adaptive scheme is less affected by noise and has better robustness and adaptability.

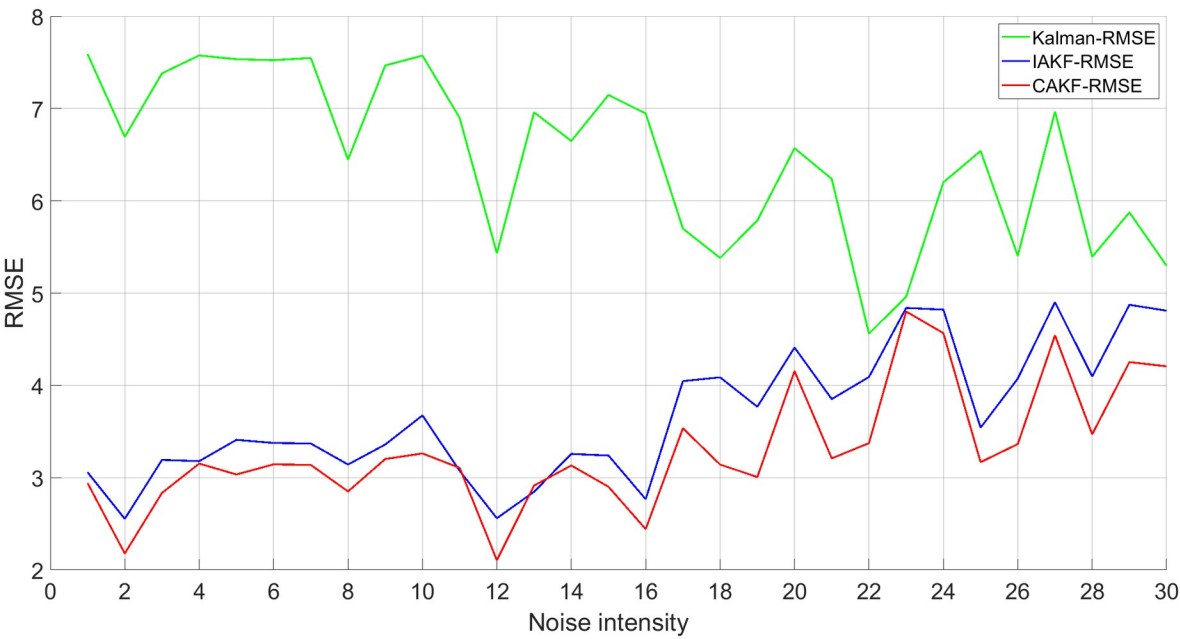

**Fig 6. Fusion effect under different Gaussian noise.**

## Experiment

This experiment is based on the technology verification line of permanent magnetic levitation rail transportation system of Jiangxi University of Science and Technology. In the experiment, INS sensor, Doppler radar sensor and GNSS sensor are respectively used to position the

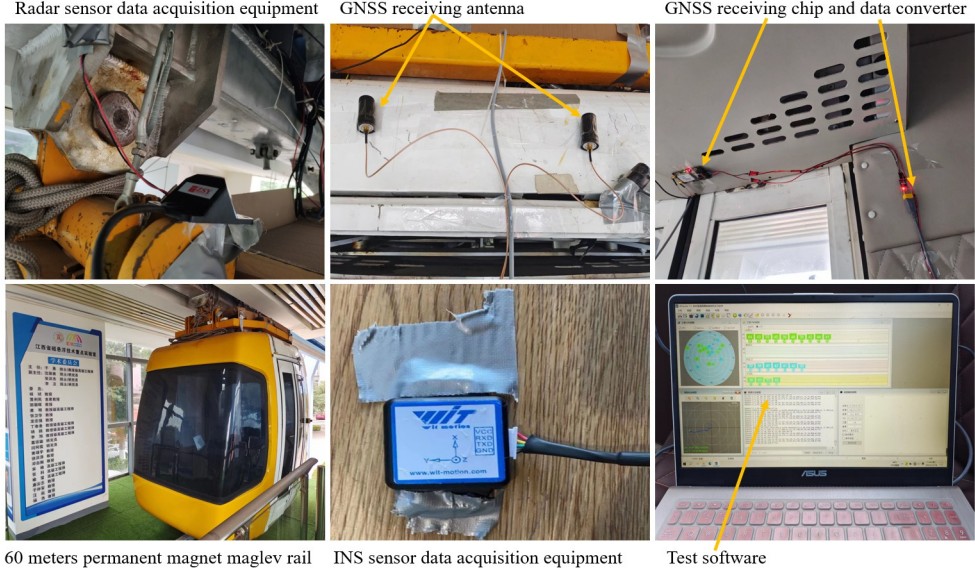

**Fig 7. Positioning test of the technology verification line of the permanent magnet maglev rail transit system.**
Reprinted from [Jiangxi University of Science and Technology Permanent Magnet Magnetic Levitation Railway Transportation System Technology Validation Line] under a CC BY license, with permission from [Jiangxi University of Science and Technology], original copyright [2019].

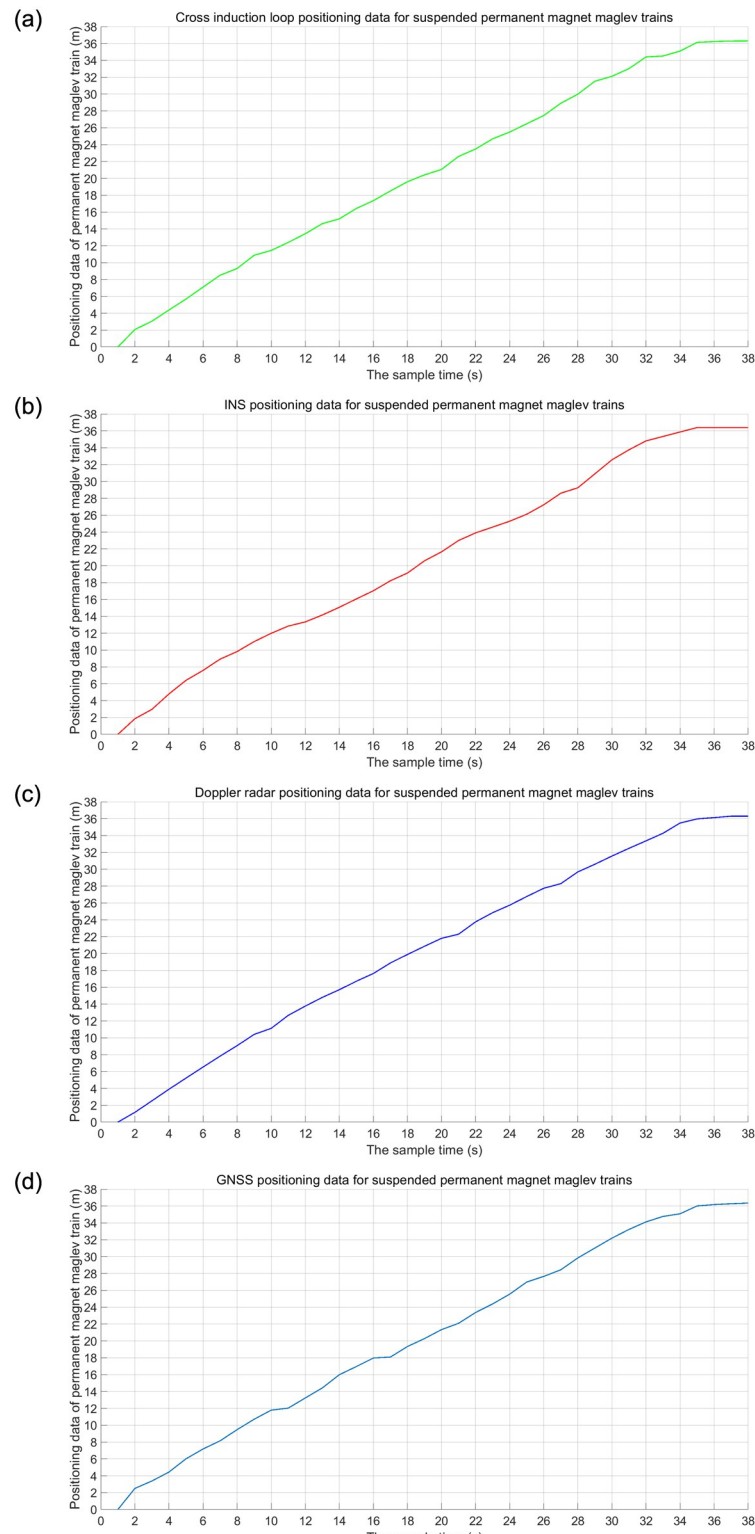

**Fig 8. Suspended rare earth permanent magnet maglev train positioning data acquisition.** (a) Data from cross induction loop; (b) Positioning data from INS; (c) Positioning data of Doppler radar; (d) Positioning data of GNSS.

suspended rare earth permanent magnet maglev train (Fig 7). The precise positioning of the suspended rare earth permanent magnet maglev train is provided by the cross induction loop.

## Experimental results of self-correcting weighted multi-sensor information fusion algorithm

The data acquisition results of INS, Doppler radar and GNSS for suspended rare earth permanent magnet maglev trains are shown in Fig 8. In the self-correcting weighted fusion algorithm, the RMSE is used as an evaluation metric to verify the effect of different $\tau$ values on the fusion results. The results in Fig 9 were obtained by calculating the values of RMSE for $\tau$ values from 0.01 to 1.

As can be seen in Fig 9, there is some error in the fusion results when $\tau$ is taken to different values. When the $\tau$ value is 0.94, the RMSE value of the fusion result with the localization value of the cross induction loop is the smallest. It is shown that the highest accuracy of self-correcting weighted multi-sensor fusion localization for suspended rare-earth permanent magnet maglev trains is achieved at $\tau = 0.94$. In order to ensure the optimal fusion effect of sensors, in the subsequent self-correcting weighted fusion algorithm, $\tau = 0.94$.

Fig 10 shows the positioning results of the suspended rare earth permanent magnet maglev train using INS, Doppler radar, GNSS and self-correcting weighted multi-sensor fusion algorithms, respectively. In order to evaluate the positioning accuracy of four methods for permanent magnet maglev trains, ME and RMSE were used as performance indicators to calculate the errors between the four methods and the cross induction loop. ME is used to measure whether the results are unbiased and is calculated by Eq (35). The calculation results are

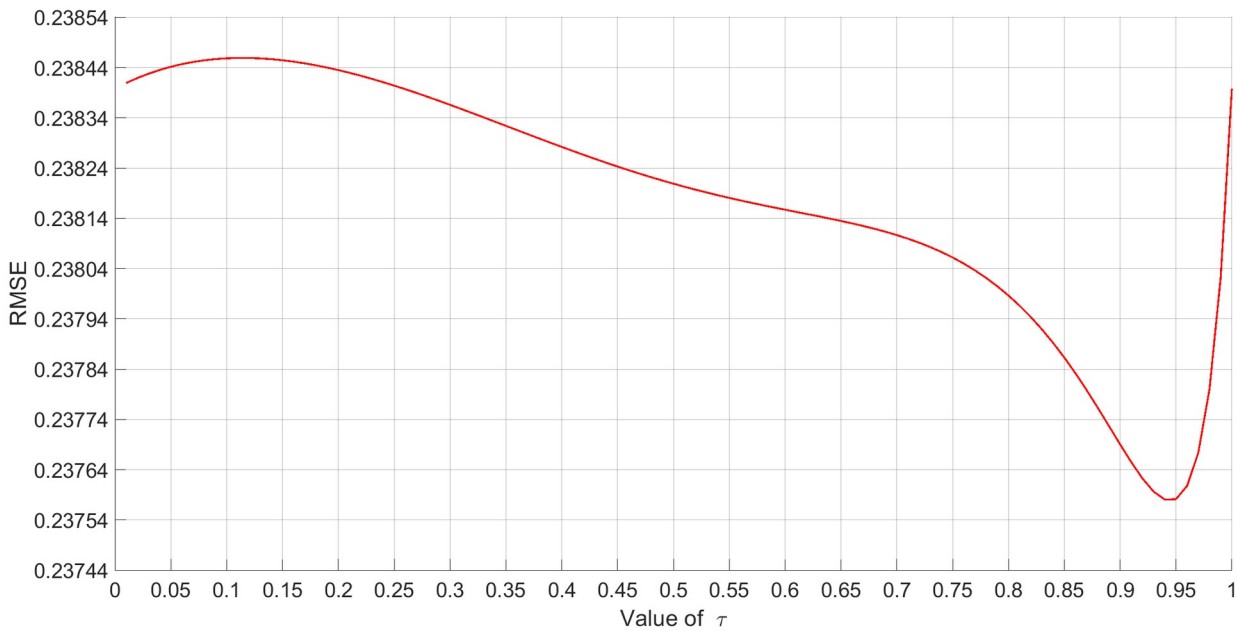

**Fig 9. Effect of different $\tau$ values on self-correcting weighted fusion algorithm.**

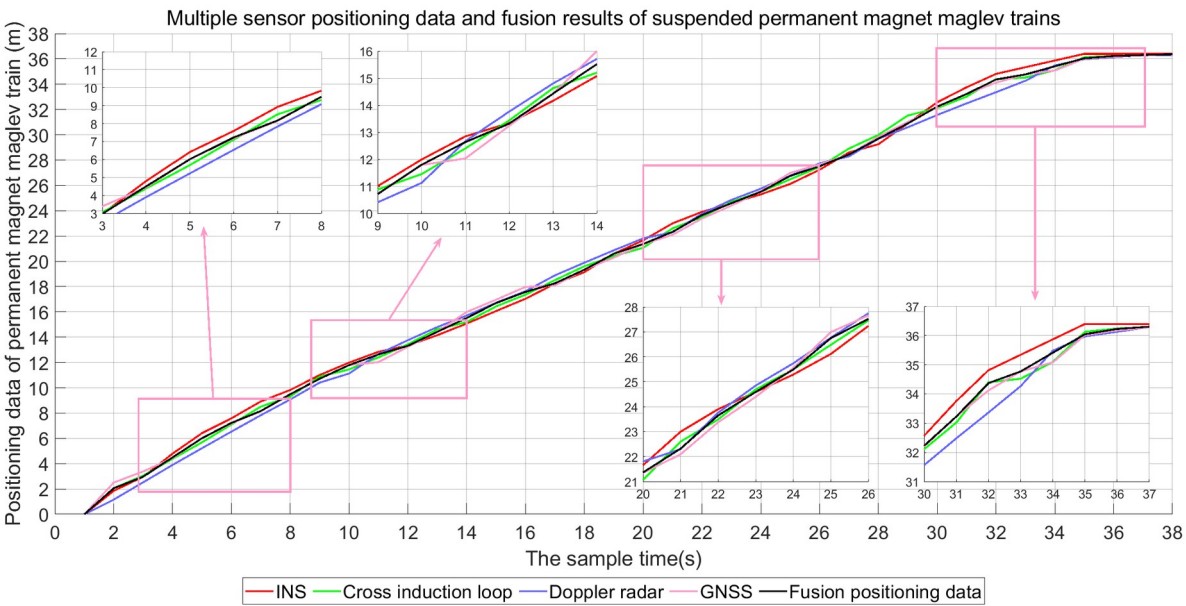

**Fig 10. Self-correcting weighted fusion algorithm fusion positioning results.**

shown in Table 1.

$$ME = \frac{\sum_{k=1}^{N}(y_k - \hat{x}_k)}{N} \tag{35}$$

Fig 10 shows that sensor fluctuations have a small effect on the values of the self-correcting weighted multi-sensor fusion algorithm. When the positioning data from INS, Doppler radar and GNSS show large differences, the self-correcting weighted multi-sensor fusion results are closer to the actual positioning data than the three sensors.

From Table 1, it can be seen that the MEs of INS, Doppler radar, GNSS, and self-correcting weighted fusion algorithms are 0.3694, 0.3894, 0.2627, and 0.1950, respectively. The RMSEs are 0.4282, 0.4582, 0.3225, and 0.2376, respectively. The ME of the self-correcting weighted fusion algorithm is reduced by 47.212%, 49.923%, and 25.771% compared to the three sensors, respectively. RMSE decreased by 44.512%, 48.145%, and 26.326%, respectively. These results all show that the self-correcting weighted multi-sensor fusion algorithm has lower errors. The positioning accuracy is closer to the cross induction loop, which is more accurate for the positioning of permanent magnet maglev trains.

**Table 1. Errors of INS, Doppler radar, GNSS and fusion algorithms.**

| Evaluation Indicators | INS | Doppler radar | GNSS | Fusion Algorithm |
|---|---|---|---|---|
| ME | 0.3694 | 0.3894 | 0.2627 | 0.1950 |
| RMSE | 0.4282 | 0.4582 | 0.3225 | 0.2376 |

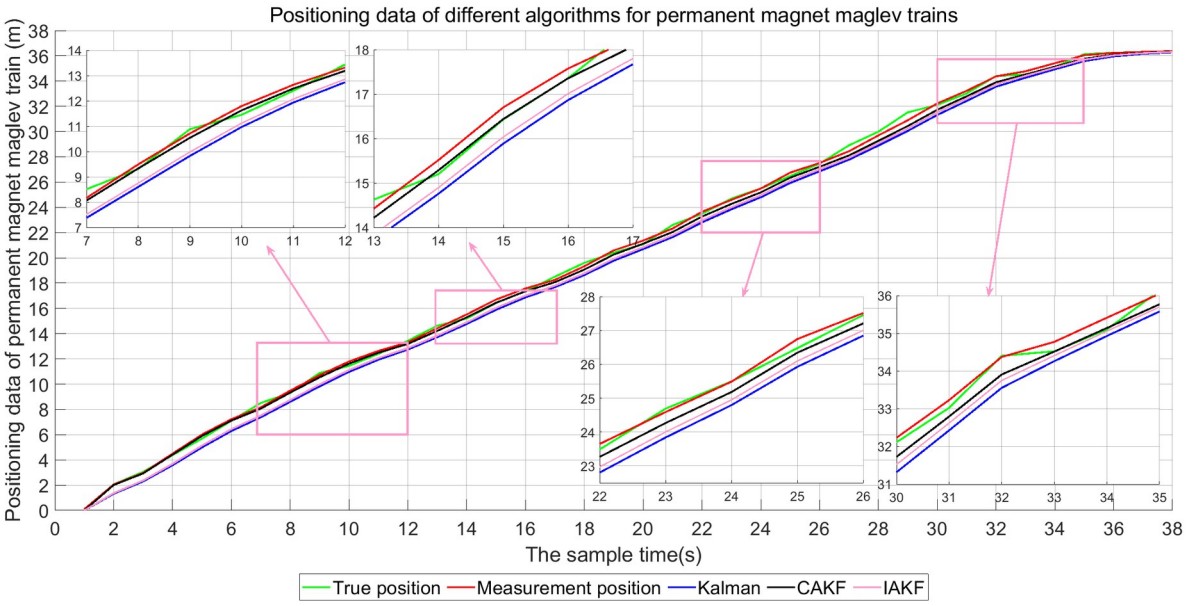

**Fig 11. Positioning effect of suspended rare earth permanent magnet maglev train.**

## Experimental results of CAKF

Next, the localization accuracy of the three algorithms, Kalman, IAKF and CAKF, for the permanent magnet maglev rail transit system is verified. The self-correcting weighted fusion results are processed using each of the three algorithms, and the results are shown in Fig 11. The positioning errors of the three algorithms are shown in Fig 12 and Table 2.

Fig 11 shows that the positioning of the suspended rare-earth permanent magnet maglev train can be achieved more accurately using CAKF. In the starting phase of localization, the localization results of Kalman algorithm and IAKF differ from the real values by about 1m, while the localization results of CAKF do not differ much from the real values. In the middle and late stages of localization, IAKF localizes better than Kalman algorithm, but there is a large gap with the true value. The localization of CAKF has a much smaller gap with the true value. This shows that CAKF always maintains effective positioning and higher positioning accuracy.

Fig 12 shows that the MEs of Kalman's algorithm are all less than 1.5 and the RMSEs are all less than 2.25. The MEs of IAKF are all less than 1.3 and the RMSEs are all less than 1.7. The MEs of CAKF are all less than 1.1 and the RMSEs are all less than 1.2. The error magnitudes of CAKF are significantly smaller than those of Kalman and IAKF. Except for the 26–30 sampling points, the ME of CAKF is less than 0.6 and the RMSE is less than 0.3, which are lower than the errors of 1.25 for Kalman and 1 for IAKF.

Fig 13 and Table 2 show that for ME error, Kalman is 0.6693, IAKF is 0.5341, and CAKF is 0.2576. For the RMSE, Kalman is 0.7384, IAKF is 0.6083, and CAKF is 0.3569. The ME of CAKF is reduced by 61.512%, 51.769% compared to the other two algorithms. The RMSE is decreased by 51.666%, 41.328%. The data in Fig 13 and Table 2 both indicate that CAKF has a smaller deviation from the actual position of the permanent magnet maglev train, making the positioning more accurate.

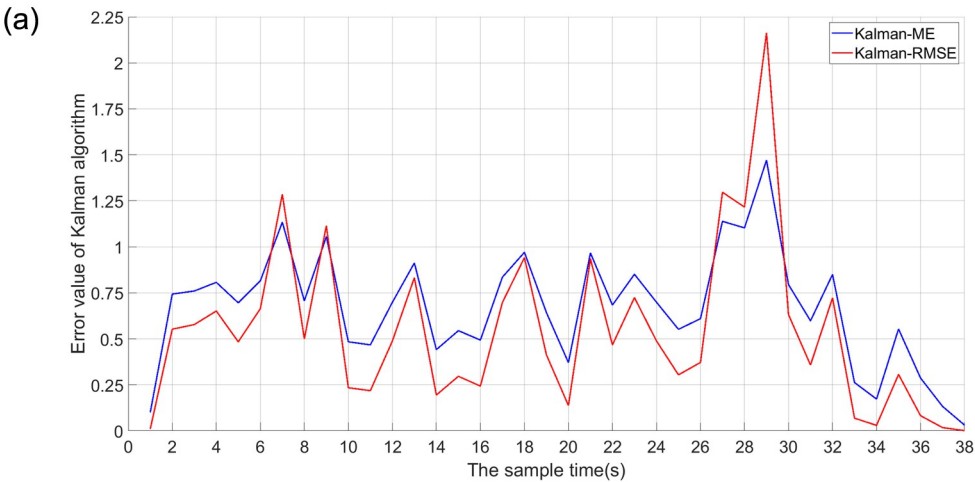

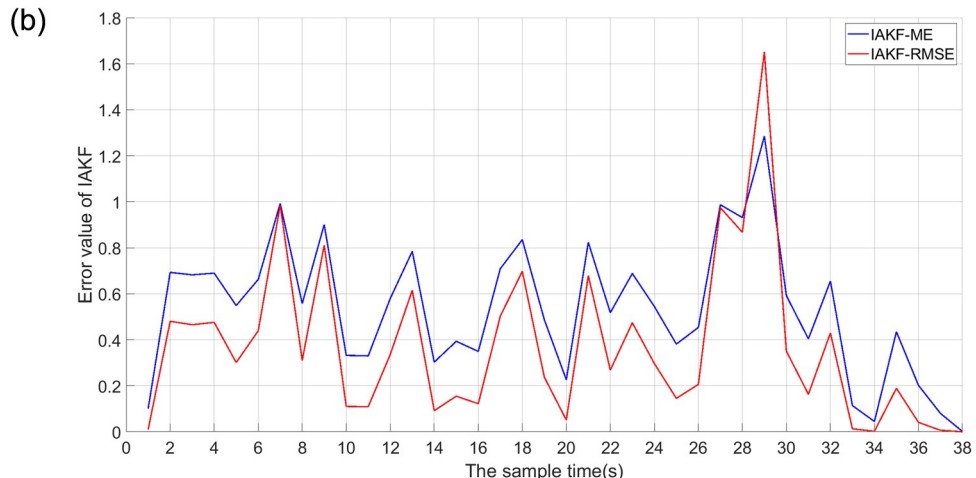

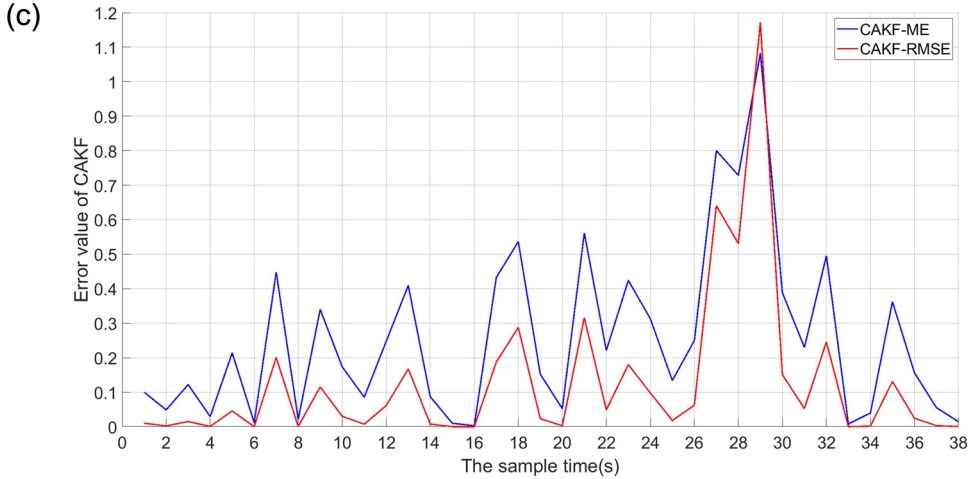

**Fig 12. Error values of the three filtering methods.** (a) Kalman's error value; (b) IAKF's error value;(c) Error value of CAKF.

**Table 2. Positioning evaluation of suspended rare earth permanent magnet magnetic levitation.**

| Evaluation Indicators | Kalman | IAKF | CAKF |
|---|---|---|---|
| ME | 0.6693 | 0.5341 | 0.2576 |
| RMSE | 0.7384 | 0.6083 | 0.3569 |

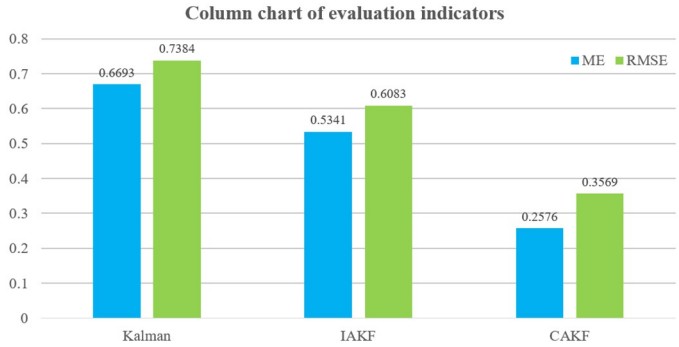

**Fig 13. Histogram of evaluation indicators.**

## Self-correcting weighted multi-sensor information fusion algorithm combined with CAKF algorithm for the localization of permanent magnet maglev trains

In order to compare the fusion positioning effect of a single sensor and multiple sensors, the suspended permanent magnet maglev train is located using INS, Doppler radar, GNSS and three sensor self-correcting weighted fusion results. The positioning results using the CAKF

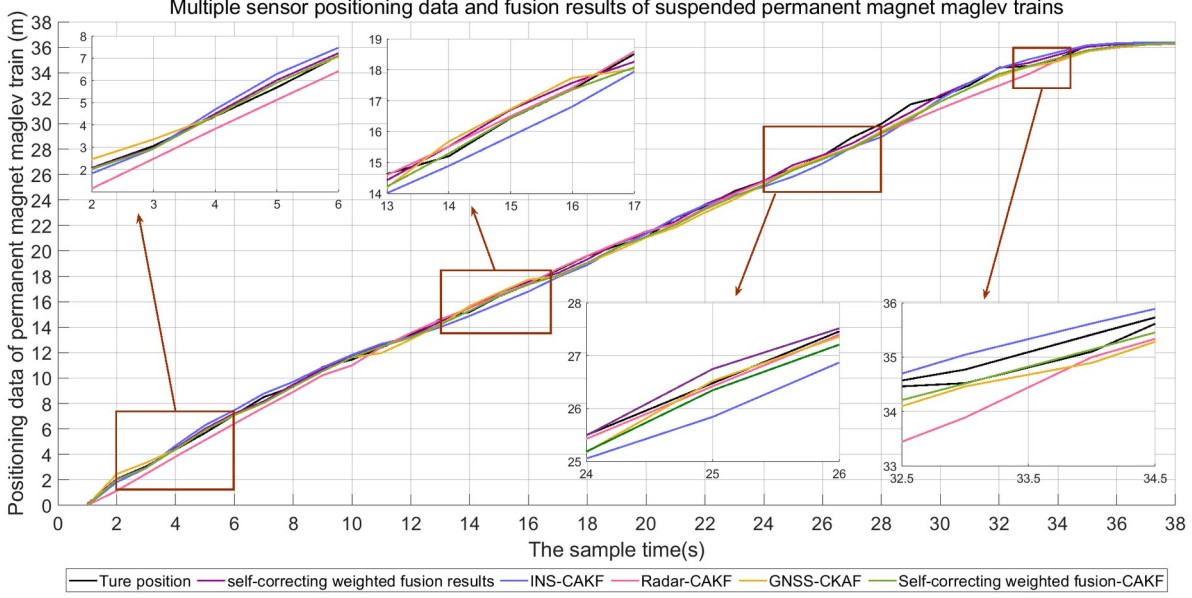

**Fig 14. Fusion results of different sensors.**

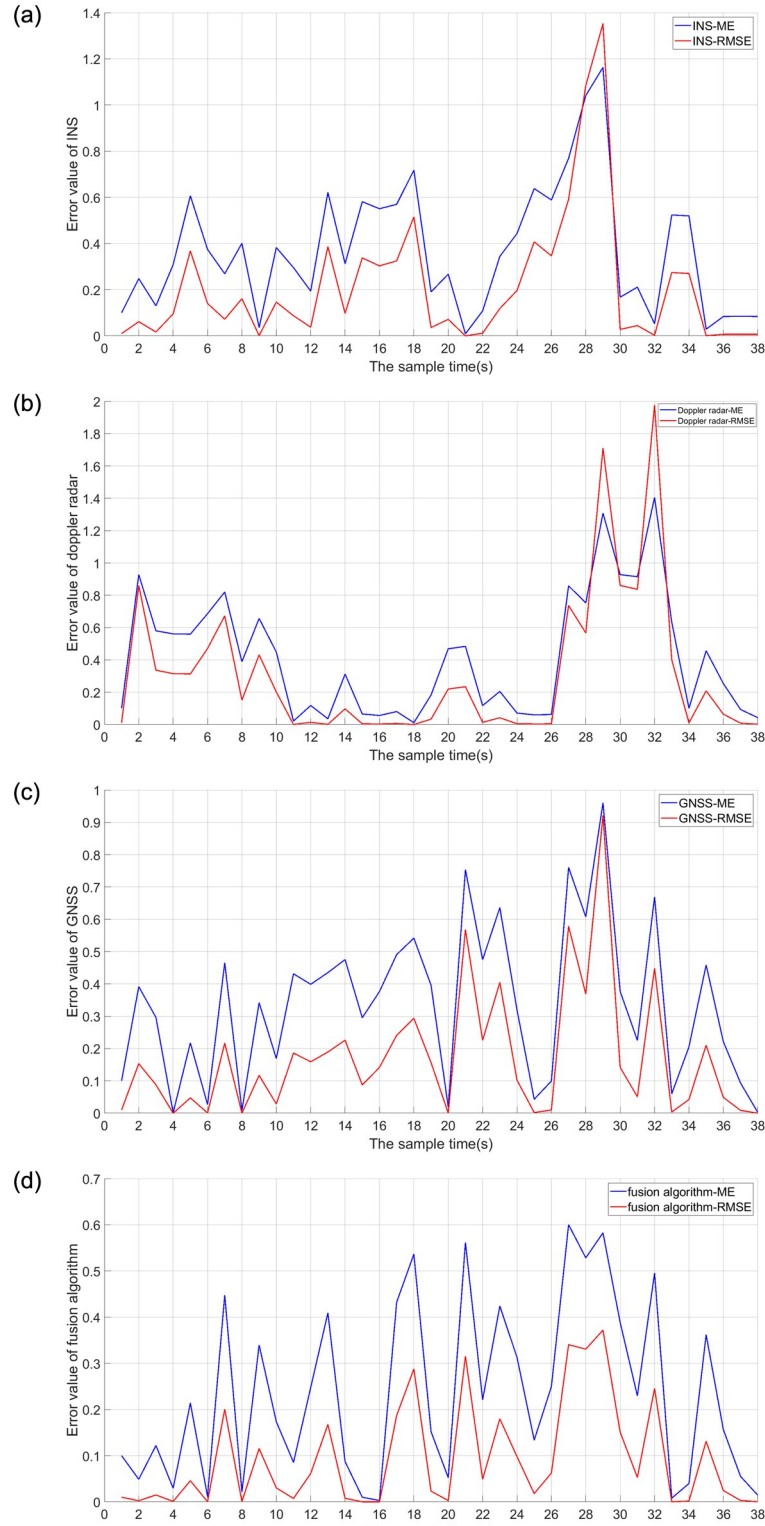

**Fig 15. Error analysis of different positioning methods.** (a) INS error results; (b) Doppler radar error results; (c) GNSS error results; (d) Error results of fusion algorithm.

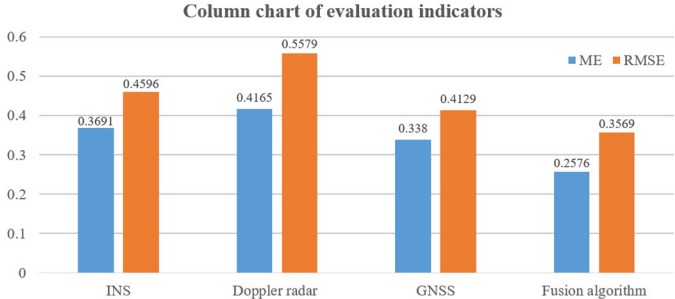

**Fig 16. Histogram of evaluation indicators.**

**Table 3. Evaluation of positioning errors of different sensors.**

| Evaluation Indicators | INS | Doppler radar | GNSS | Proposed algorithm |
|---|---|---|---|---|
| ME | 0.3691 | 0.4165 | 0.3380 | 0.2576 |
| RMSE | 0.4593 | 0.5579 | 0.4129 | 0.3569 |

are shown in Fig 14. The error of comparing the positioning results using four methods with the cross induction loop is shown in Fig 15.

As can be seen in Figs 14 and 15, the localization using the self-correcting weighted multi-sensor fusion algorithm is better than the localization using a single INS, Doppler radar, and GNSS. ME error stays below 0.6 and RMSE stays below 0.4. The fluctuations are smaller and closer to the actual position.

From Fig 16 and Table 3, it can be seen that the ME and RMSE of the suspended rare earth permanent magnet maglev train using INS are 0.3691 and 0.4596, respectively; Doppler radar is 0.4165 and 0.5579, respectively; GNSS is 0.3380 and 0.4129, respectively; and proposed algorithm is 0.2576 and 0.3569, respectively. The ME and RMSE of the proposed algorithm were reduced by 30.209% and 22.346% compared to INS; by 38.151% and 36.028% compared to Doppler radar; and by 23.787% and 13.563% compared to GNSS. It is demonstrated that the localization position using multiple sensors has a smaller deviation from the actual position of permanent magnet maglev trains and is closer to the actual value than using a single sensor. This indicates that proposed algorithm has better localization effect and localization accuracy in fusion localization.

## Conclusion

A self-correcting weighted multi-sensor fusion algorithm is proposed to solve the problem of biased fusion results due to large errors in multi-sensor localization of levitated rare earth permanent magnet maglev trains. For the Kalman filtering problem without accurate statistical process noise, a new adaptive Kalman filter is used to relax the key constraint of Kalman theory on the process noise covariance Q by using a feedback adaptation of the a posteriori sequence to the a priori error covariance. The experimental results of the suspended rare earth permanent magnet maglev train show that the RMSE is reduced by 44.512%, 48.145%, and 26.326% after fusing three sensors, INS, Doppler radar, and GNSS, respectively, using the self-correcting weighted multi-sensor fusion algorithm. Compared to Kalman and IAKF, the RMSE is reduced by 51.666% and 41.328% using CAKF, respectively. The RMSE is reduced by 22.346%, 36.028% and 13.563% using the fusion algorithm compared to the CAKF algorithm using a single sensor,

respectively. Simulation and experimental results demonstrate that the self-correcting weighted multi-sensor fusion algorithm and the adaptive Kalman algorithm are outstanding in terms of fault tolerance performance, filtering adaptability and accuracy, and can meet the requirements for the positioning of suspended rare earth permanent magnet maglev trains in practice.

## Supporting information

**S1 Table. INS, cross sensing loop, Doppler radar, and GNSS positioning data.**
(XLSX)

## Author Contributions

**Conceptualization:** Yiwei Xu, Kuangang Fan.

**Data curation:** Yiwei Xu, Qian Hu.

**Funding acquisition:** Kuangang Fan.

**Methodology:** Yiwei Xu.

**Writing – original draft:** Yiwei Xu.

**Writing – review & editing:** Kuangang Fan, Qian Hu, Haoqi Guo.

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
