## [Decision Letter · Decision Letter 0]

31 Jul 2023

PONE-D-23-18983Multi-sensor information fusion localization of rare-earth suspended permanent magnet maglev trains based on adaptive kalman algorithm.PLOS ONE

Dear Dr. Fan,

Thank you for submitting your manuscript to PLOS ONE. After careful consideration, we feel that it has merit but does not fully meet PLOS ONE’s publication criteria as it currently stands. Therefore, we invite you to submit a revised version of the manuscript that addresses the points raised during the review process.

We look forward to receiving your revised manuscript.

Kind regards,

Salim Heddam

Academic Editor

PLOS ONE

5. We note that Figures 1 and 7 in your submission contain copyrighted images. All PLOS content is published under the Creative Commons Attribution License (CC BY 4.0), which means that the manuscript, images, and Supporting Information files will be freely available online, and any third party is permitted to access, download, copy, distribute, and use these materials in any way, even commercially, with proper attribution. For more information, see our copyright guidelines: http://journals.plos.org/plosone/s/licenses-and-copyright.

1. You may seek permission from the original copyright holder of Figures 1 and 7 to publish the content specifically under the CC BY 4.0 license.

Additional Editor Comments:

Reviewer 1:This article proposes a self-correcting weighted fusion estimation algorithm for localization of rare-earth suspended permanent magnet maglev trains. The main issues are listed as follows:

1、The authors need to provide a brief introduction to the purpose and importance of maglev train positioning, as many readers may not be familiar with maglev trains but are interested in multi-sensor fusion.

2、Why the random weighting method is introduced in lines 25-31, and whether this method is compared with the mentioned method in the experiment.

3、In line 33, it is mentioned that the Kalman filter can be used for navigation, target tracking, and control. It is suggested to include a reference citation after "target tracking," such as:

*Object tracking in satellite videos: correlation particle filter tracking method with motion estimation by Kalman filter. IEEE Transactions on Geoscience and Remote Sensing. 2022.

*Learning cross-attention discriminators via alternating time-space transformers for visual tracking. IEEE Transactions on Neural Networks and Learning Systems. 2023.

4、The authors mentioned that a 60-meter-long permanent magnet magnetic levitation rail transit system technology verification line was completed, where the positioning method used is cross induction loop, which was introduced before 2013. Over such a long period of time, there should have been research and proposals for alternative solutions to the cross induction loop. It is necessary for the authors to introduce this topic with appropriate elaboration, as I believe it is crucial for introducing the methodology of this paper.

5、In line 77, it is mentioned that although the centralized fusion algorithm has higher accuracy than the distributed fusion algorithm, however, it is prone to the problem of dimensional explosion when the number of sensors is large. In the experiments conducted in this paper, how many sensors did the authors use? Did the authors employ a centralized fusion algorithm in the experiments, and if so, what were the results?

Reviewer 2:

1.The references in 2023 are not available, and the latest references are added.

2. Magnet maglev train speed is fast, how to process information quickly, how to calculate the realization.

3. The font of axis in Figure 4-15 is too small to read clearly.

4.The estimation of covariance Q and covariance R by adaptive Kalman filter is relatively mature. In this paper, only the estimation of Q is not convincing enough.

5. In Figure 7, no processor is seen. What processing method is adopted after data collection? Please specify.

6. In the review, the application of Kalman, the development of the suspension vehicle and the application of multiple sensors are described. The application of Kalman in the maglev vehicle is missing, isn't it?

7. ME and EMSE are used in this paper for evaluation. Can you specify the actual error data, the actual position and the position gap using this method?

Reviewers' comments:

Reviewer's Responses to Questions

**Comments to the Author**

1. Is the manuscript technically sound, and do the data support the conclusions?

Reviewer #1: Yes

Reviewer #2: Yes

2. Has the statistical analysis been performed appropriately and rigorously? 

Reviewer #1: Yes

Reviewer #2: Yes

3. Have the authors made all data underlying the findings in their manuscript fully available?

Reviewer #1: Yes

Reviewer #2: Yes

4. Is the manuscript presented in an intelligible fashion and written in standard English?

Reviewer #1: Yes

Reviewer #2: Yes

5. Review Comments to the Author

Reviewer #1: This article proposes a self-correcting weighted fusion estimation algorithm for localization of rare-earth suspended permanent magnet maglev trains. The main issues are listed as follows:

1、The authors need to provide a brief introduction to the purpose and importance of maglev train positioning, as many readers may not be familiar with maglev trains but are interested in multi-sensor fusion.

2、Why the random weighting method is introduced in lines 25-31, and whether this method is compared with the mentioned method in the experiment.

3、In line 33, it is mentioned that the Kalman filter can be used for navigation, target tracking, and control. It is suggested to include a reference citation after "target tracking," such as:

*Object tracking in satellite videos: correlation particle filter tracking method with motion estimation by Kalman filter. IEEE Transactions on Geoscience and Remote Sensing. 2022.

*Learning cross-attention discriminators via alternating time-space transformers for visual tracking. IEEE Transactions on Neural Networks and Learning Systems. 2023.

4、The authors mentioned that a 60-meter-long permanent magnet magnetic levitation rail transit system technology verification line was completed, where the positioning method used is cross induction loop, which was introduced before 2013. Over such a long period of time, there should have been research and proposals for alternative solutions to the cross induction loop. It is necessary for the authors to introduce this topic with appropriate elaboration, as I believe it is crucial for introducing the methodology of this paper.

5、In line 77, it is mentioned that although the centralized fusion algorithm has higher accuracy than the distributed fusion algorithm, however, it is prone to the problem of dimensional explosion when the number of sensors is large. In the experiments conducted in this paper, how many sensors did the authors use? Did the authors employ a centralized fusion algorithm in the experiments, and if so, what were the results?

Reviewer #2: 1.The references in 2023 are not available, and the latest references are added.

2. Magnet maglev train speed is fast, how to process information quickly, how to calculate the realization.

3. The font of axis in Figure 4-15 is too small to read clearly.

4.The estimation of covariance Q and covariance R by adaptive Kalman filter is relatively mature. In this paper, only the estimation of Q is not convincing enough.

5. In Figure 7, no processor is seen. What processing method is adopted after data collection? Please specify.

6. In the review, the application of Kalman, the development of the suspension vehicle and the application of multiple sensors are described. The application of Kalman in the maglev vehicle is missing, isn't it?

7. ME and EMSE are used in this paper for evaluation. Can you specify the actual error data, the actual position and the position gap using this method?

6. PLOS authors have the option to publish the peer review history of their article (what does this mean?). If published, this will include your full peer review and any attached files.

Reviewer #1: No

Reviewer #2: No

---

## [Author Response · Author response to Decision Letter 0]

8 Sep 2023

Dear Reviewers:

Thanks for your letter and the reviewers’ comments on our manuscript entitled “Multi-sensor information fusion localization of rare-earth suspended permanent magnet maglev trains based on adaptive Kalman algorithm” (ID: PONE-D-23-18983R1). These comments are all valuable and very helpful for revising and improving our paper, as well as have important guiding significance for our research. We have carefully considered all comments from the reviewers and revised our manuscript accordingly. Below are our individual responses to each of the reviewers' comments. We believe that our responses have well addressed all concerns from the reviewers. We hope our revised manuscript can be accepted for publication. We have also attached a response to the editor and two reviewers, in which we have responded to the comments made by the editor and reviewers (details of the changes are presented in Response to Reviewers).

Responds to the Reviewers' comments：

Reviewer 1: This article proposes a self-correcting weighted fusion estimation algorithm for localization of rare-earth suspended permanent magnet maglev trains. The main issues are listed as follows:

Q1: The authors need to provide a brief introduction to the purpose and importance of maglev train positioning, as many readers may not be familiar with maglev trains but are interested in multi-sensor fusion.

A1: We thank the reviewer for reading our paper carefully and giving the above comment. In lines 11-17, we describe the purpose and importance of positioning the Maglev train. The revisions are highlighted in red.

Q2: Why the random weighting method is introduced in lines 25-31, and whether this method is compared with the mentioned method in the experiment.

A2: In lines 28-32 of the paper, it is described in detail that multi-sensors may suffer from data error or even distortion by noise or environmental interference in practical applications. Therefore, a weighted fusion algorithm is used to solve this problem. In lines 33-40, this paper adopts a weighted approach to solve the limitations of multi-sensor systems and proposes a new weight calculation algorithm, namely the self-correcting weighted multi-sensor fusion algorithm. Through the introduction of the decay memory factor, the calculation of the weights is improved. The weights of each sensor are calculated based on the calculated deviation of each sensor measurement from the center point and the sum of the deviation squares to get more accurate results.

The Random Weighted Algorithm is a type of weighted algorithm. The original manuscript only described the random weighted algorithm, which was inaccurately stated and is now corrected. The revisions are highlighted in red.

Q3: In line 33, it is mentioned that the Kalman filter can be used for navigation, target tracking, and control. It is suggested to include a reference citation after "target tracking," such as:

*Object tracking in satellite videos: correlation particle filter tracking method with motion estimation by Kalman filter. IEEE Transactions on Geoscience and Remote Sensing. 2022.

*Learning cross-attention discriminators via alternating time-space transformers for visual tracking. IEEE Transactions on Neural Networks and Learning Systems. 2023.

A3: Thanks for your valuable suggestions. In lines 37-38, we add references [21], [22]. The revisions are highlighted in red.

Q4: The authors mentioned that a 60-meter-long permanent magnet magnetic levitation rail transit system technology verification line was completed, where the positioning method used is cross induction loop, which was introduced before 2013. Over such a long period of time, there should have been research and proposals for alternative solutions to the cross induction loop. It is necessary for the authors to introduce this topic with appropriate elaboration, as I believe it is crucial for introducing the methodology of this paper.

A4: The positioning method currently used for the permanent magnetic levitation rail system technology validation line is the cross induction loop. The cross induction loop is a relative speed measurement and positioning means with high positioning accuracy. However, it has the disadvantages of complex equipment and instruments, installation difficulties, high maintenance requirements, and high cost. Especially, the cost of cross induction loop is in the range of 3-5 million yuan per kilometer, which is very high, and this is not conducive to the popularization and application of permanent magnetic levitation rail transit system.

Thank you for your suggestion, and the subject of cross induction loop alternatives has been appropriately addressed on lines 71-78. The revisions are highlighted in red.

Q5: In line 77, it is mentioned that although the centralized fusion algorithm has higher accuracy than the distributed fusion algorithm, however, it is prone to the problem of dimensional explosion when the number of sensors is large. In the experiments conducted in this paper, how many sensors did the authors use? Did the authors employ a centralized fusion algorithm in the experiments, and if so, what were the results?

A5: The centralized fusion algorithm mainly combines the multi-sensor observation equations into one expanded observation equation, and then associates them with the state equations to obtain the corresponding centralized estimates. The distributed fusion algorithm first solves the local estimates of each sensor, and then weights the local estimates to obtain the final distributed estimates. Since the centralized fusion algorithm is the projection of the state quantities on the linear space composed of the expanded dimensional observation vectors, it is the globally optimal estimation, and the centralized fusion algorithm is more accurate than the distributed fusion algorithm. However, the centralized fusion method expands the dimension of the observation equations, which makes the observation vectors higher dimensional, computationally intensive, and poorly fault-tolerant.

In the experiments of this paper, four sensors are used, which are cross induction loop, INS, Doppler radar and GNSS. Among them, due to the high positioning accuracy of the cross induction loop, the positioning data of the cross induction loop is used as the real value, and the method proposed in this article is used to fuse the INS, Doppler radar, and GNSS sensor data with multi-sensor data. Considering that the real-time and accuracy requirements need to be fulfilled in real experiments for the positioning of suspended rare earth permanent magnet maglev trains. Due to the higher computational burden of centralized fusion algorithms compared to distributed fusion algorithms, they require more time for computation and have poorer fault tolerance compared to distributed fusion methods. Therefore, in this experiment, only distributed fusion algorithms were used.

Reviewer 2:

Q1: The references in 2023 are not available, and the latest references are added.

A1: Thanks for pointing out this issue in our manuscript. We have revised the manuscript according to your suggestions. In lines 9, we reintroduce references [5] and [6].

Q2: Magnet maglev train speed is fast, how to process information quickly, how to calculate the realization.

A2: Rare earth permanent magnet magnetic levitation rail transit system is a low to medium speed, small capacity and intelligent transportation mode which can adapt to complex terrain. In this experiment, the cross induction loop, INS, Doppler radar and GNSS sensors are set up to collect real-time data through the sensors, and then the information fusion of multi-sensor data is carried out using the method proposed in this paper to realize the localization of suspended rare-earth permanent magnet magnetic levitation trains. The computation time of the proposed algorithm is 0.2 seconds, and the method proposed in this paper can meet the positioning requirements of the rare-earth permanent magnet magnetic levitation rail transit system.

Q3: The font of axis in Figure 4-15 is too small to read clearly.

A3: Thanks for pointing out this issue in our manuscript. We have revised the manuscript according to your suggestions. We've enlarged the font for the axes in Figure 4-15.

Q4: The estimation of covariance Q and covariance R by adaptive Kalman filter is relatively mature. In this paper, only the estimation of Q is not convincing enough.

A4: Thank you for asking this question, our research at this stage focuses on relaxing the critical constraints of Kalman theory on the process noise covariance Q, and proposes a covariance prediction scheme that utilizes a posteriori sequences to feedback adaptively to the a priori error covariance. The study of the covariance R is the next stage of our research, and we will subsequently enhance our study and research on the estimation of the covariance Q and covariance R.

Q5: In Figure 7, no processor is seen. What processing method is adopted after data collection? Please specify.

A5: In this paper, four sensors are used in the experiments, which are cross induction loop, INS, Doppler radar and GNSS. among them, due to the high positioning accuracy of cross induction loop, the positioning data of cross induction loop is taken as the real value, and the three types of sensors' data, which are INS, Doppler radar and GNSS, are fused to the multi-sensor data using the method proposed in this paper.

In the experiment, the data acquisition interval of the four types of sensors is set to be 1 second, and the experiment time is 38 seconds. After the data collection of the experiment, the four types of sensor positioning data are transformed into position data respectively using MATLAB, and then the multi-sensor information fusion is carried out to realize the positioning of the permanent magnet magnetic levitation train. The experimental equipment and field environment are shown in Response to Reviewers Fig1.

Q6: In the review, the application of Kalman, the development of the suspension vehicle and the application of multiple sensors are described. The application of Kalman in the maglev vehicle is missing, isn't it?

A6: According to the reviewer’s suggestion, we introduce the application of Kalman technique to Maglev trains. we cite references [34]-[36] and then briefly describe them.

Q7: ME and EMSE are used in this paper for evaluation. Can you specify the actual error data, the actual position and the position gap using this method?

A7: In this paper, INS, GNSS and Doppler radar are used to localize the suspended permanent magnet maglev train, and the results of the three localization methods are fused with the information of the algorithm proposed in this paper to realize the permanent magnet maglev train localization based on the fusion of multi-sensor information. In order to check the positioning accuracy of the proposed algorithm, this paper compares the cross induction loop data as actual values with the proposed algorithm positioning results, and uses ME and RMSE as two error assessment methods to obtain the error results.

 From Fig. 2, Fig. 3 and Table 1 in Response to Reviewwes, it can be seen that the ME of INS is lower than 1.2, and in most of the time, the ME is lower than 0.7.The ME of Doppler radar is lower than 1.4, and in most of the time, the ME is lower than 1. The ME of GNSS is lower than 1, and in most of the time, the ME is lower than 0.7.The ME of the proposed algorithm is lower than 0.6. the RMSE of INS is less than 1.4, and in most of the time, the RMSE is less than 0.5. the RMSE of Doppler radar is less than 2, and in most of the time, the RMSE is less than 0.9. the RMSE of GNSS is less than 1, and in most of the time, the RMSE is less than 0.6. the RMSE of the proposed algorithm is less than 0.4.

The ME and RMSE of the levitated rare earth permanent magnet maglev train using INS are 0.3691 and 0.4596, respectively; Doppler radar is 0.4165 and 0.5579, respectively; GNSS is 0.3380 and 0.4129, respectively; and the proposed algorithm is 0.2576 and 0.3569, respectively.

From the ME and RMSE of the four localization means, it can be seen that the deviation between the localized position and the actual position of the PME train using multiple sensors is smaller than that using a single sensor, and the localization results of the algorithm proposed in this paper are closer to the actual position of the PME train, with a smaller positional gap closer to the actual value. This shows that the fusion algorithm proposed in this paper has better localization results and accuracy in fusion localization.

Thanks for the above comments. We have revised them according to your suggestions. We hope our revised manuscript can be accepted for publication.

---

## [Decision Letter · Decision Letter 1]

18 Sep 2023

Multi-sensor information fusion localization of rare-earth suspended permanent magnet maglev trains based on adaptive kalman algorithm

PONE-D-23-18983R1

Dear Dr. Fan

We’re pleased to inform you that your manuscript has been judged scientifically suitable for publication and will be formally accepted for publication once it meets all outstanding technical requirements.

Kind regards,

Salim Heddam

Academic Editor

PLOS ONE

Additional Editor Comments (optional):

Reviewer 1:The author has effectively addressed the concerns I raised in my initial review. Based on the revisions made and the overall quality of the paper, I am in agreement with accepting this manuscript for publication.

Reviewer 2:Thank you very much for the author's reply, I have carefully reviewed them.In my opinion, the manuscript basically meets the requirements of the journal.

Reviewers' comments:

Reviewer's Responses to Questions

**Comments to the Author**

1. If the authors have adequately addressed your comments raised in a previous round of review and you feel that this manuscript is now acceptable for publication, you may indicate that here to bypass the “Comments to the Author” section, enter your conflict of interest statement in the “Confidential to Editor” section, and submit your "Accept" recommendation.

Reviewer #1: All comments have been addressed

Reviewer #2: All comments have been addressed

2. Is the manuscript technically sound, and do the data support the conclusions?

Reviewer #1: Yes

Reviewer #2: Yes

3. Has the statistical analysis been performed appropriately and rigorously? 

Reviewer #1: Yes

Reviewer #2: Yes

4. Have the authors made all data underlying the findings in their manuscript fully available?

Reviewer #1: Yes

Reviewer #2: Yes

5. Is the manuscript presented in an intelligible fashion and written in standard English?

Reviewer #1: Yes

Reviewer #2: Yes

6. Review Comments to the Author

Reviewer #1: The author has effectively addressed the concerns I raised in my initial review. Based on the revisions made and the overall quality of the paper, I am in agreement with accepting this manuscript for publication.

Reviewer #2: Thank you very much for the author's reply, I have carefully reviewed them.In my opinion, the manuscript basically meets the requirements of the journal.

7. PLOS authors have the option to publish the peer review history of their article (what does this mean?). If published, this will include your full peer review and any attached files.

Reviewer #1: No

Reviewer #2: No

---

## [Editor Report · Acceptance letter]

5 Oct 2023

PONE-D-23-18983R1 

Multi-sensor information fusion localization of rare-earth suspended permanent magnet maglev trains based on adaptive kalman algorithm 

Dear Dr. Fan:

I'm pleased to inform you that your manuscript has been deemed suitable for publication in PLOS ONE. Congratulations! Your manuscript is now with our production department. 

Kind regards, 

on behalf of

Dr. Salim Heddam 

Academic Editor

PLOS ONE